# Effects of increasing molybdenum under low nitrogen input on yield, nitrogen-metabolizing enzymes and nitrogen use efficiency of winter wheat cultivars

Di Yang[1,2]*, Qixia Wu[2], Youning Wang [1]*

**1** Hubei Key Laboratory of Quality Control of Characteristic Fruits and Vegetables, Hubei Engineering University, Xiaogan, China, **2** College of Agriculture, Yangtze University, Jingzhou, China

* king_union@126.com

## Abstract

In wheat production, improving resource utilization and grain yield has been a longstanding goal that researchers have been pursuing. This study aimed to investigate whether regulated nitrogen (N) and molybdenum (Mo) fertilizer management could enhance wheat yield and nitrogen use efficiency (NUE). This study reports the effects of three N application levels (N0: 0 kg N ha$^{-1}$, N1: 75 kg N ha$^{-1}$, N2: 150 kg N ha$^{-1}$) and a combination of three Mo application levels (Mo1: 0 kg Na$_2$MoO$_4$ ha$^{-1}$, Mo2: 0.75 kg Na$_2$MoO$_4$ ha$^{-1}$, Mo3: 1.5 kg Na$_2$MoO$_4$ ha$^{-1}$) on N metabolism, NUE, and yield in wheat. The results showed that average grain yield increased by 61.78% under the different N input, and 11.71% under the different Mo input levels. Nitrogen agronomic efficiency (NAE), nitrogen recovery efficiency (NRE) and partial factor productivity (PEPN) significantly (P < 0.01) increased at the low N rates (N1) but significantly (P < 0.05) decreased at high N rates (N2). NAE and PEPN were significantly (P < 0.01) different among different Mo input levels. NAE significantly decreased with the increasing Mo input. Conversely, PEPN increased with increaseing Mo input, the N1 + Mo3 had the highest PEPN.With the increase of Mo, the leaf area index and net photosynthetic rate increased significantly at all growth stages. Compared with Mo1, the leaf area index (LAI) of Mo2 and Mo3 increased by 13.68% and 19.33%, respectively, and the photosynthetic rate increased by 26.47% and 45.07%. Compared with Mo0(control), NR, GS and GOGAT increased by 1.15%, 4.60%, and 1.95% in N1 and by 3.80%, 7.31%, and 4.94% in N2, respectively. We conclude that high Mo and low nitrogen input could prolong the duration of green leaves, enhance the activity of nitrogen metabolism enzymes during the middle and late growth stages, and improve spikelets number per spike and N metabolism, thereby having potential to increase wheat yield and NUE.

**Data availability statement:** All relevant data are within the paper and its Supporting Information files.

**Funding:** This work was supported by the Natural Science Funds of Hubei Province of China (2024AFB1015). The funders had no role in study design, data collection and analysis, decision to publish, or preparation of the manuscript.

**Competing interests:** The authors have declared that no competing interests exist.

## Introduction

Wheat one of the most important crops worldwide and provides nutritive food to more than one-third of the global population, and plays a pivotal role in food industry [1]. It is our national condition to have a large population and less land. The fundamental way to solve our food security and meet people's increasing demand for wheat products is to constantly increase the yield of wheat [2].

Molybdenum is a trace element in soil. The content of molybdenum in soil of countries around the world is generally 0.5~5.0 mg/kg, and the average content is 2.3 mg/kg [3]. Molybdenum deficiency is common in soils worldwide, especially in acidic soils [4]. Despite the small demand of plants for molybdenum, molybdenum deficiency has been reported in more than 40 species of higher plants worldwide [5]. The leaf color of wheat seedlings was white when molybdenum was deficient, and severe necrosis occurred from the tip along the leaf direction [6]. Degreenization and yellowing occurred at the edge of wheat old leaves when molybdenum was deficient [7]. When soil available Mo level was insufficient, wheat dry matter accumulation was reduced, tillering was blocked, growth was slowed, and flowering was delayed [8]. Application of molybdenum fertilizer can increase tillering number of wheat, accelerate wheat growth rate, and promote early maturation of wheat [7]. Mo application can improve seedling vigor and promote growth and development of wheat [5]. Application of molybdenum fertilizer can improve the stability and integrity of organelles and membrane structure of winter wheat leaf cell, enhance cold resistance of winter wheat, and promote growth and development of winter wheat [9–10]. Molybdenum had significant effects on the nodulation and growth of artificially induced wheat [11]. With the increase of molybdenum application amount, wheat dry matter weight showed a trend of first increasing and then decreasing. High molybdenum content in wheat seeds can reduce the amount of molybdenum fertilizer applied in acidic soils. Hu et al.[12] found that molybdenum promoted the distribution of nitrogen absorbed by winter wheat to the growth center at jointing and heading stages, and nitrate reduction was more intense under balanced molybdenum and nitrogen supply. Therefore, appropriate amount of molybdenum fertilizer can be used to regulate the transport and distribution of nitrogen in plants.

The average nitrogen application amount of wheat in the North China plain had reached 424 kg ha$^{-1}$, which was much higher than the nutrient consumption in the same period [13–14]. At present, many experts have conducted studies on reduced nitrogen application. Liang et al.[15] found that there was no significant change in corn yield after reduced nitrogen application by 15%, but the highest yield was obtained by reducing nitrogen by 25%. At present, the nitrogen utilization rate of wheat production in China is only 21.2%~35.9%, which is much lower than the foreign level of 40%~60% [16]. Rational reduction of nitrogen fertilizer has become a hot topic in wheat fertilization research. Previous studies have shown that appropriate nitrogen reduction based on the amount of nitrogen applied by farmers will not reduce wheat yield [17–18].

Within a certain range of nitrogen application rate, wheat grain yield increased with the increase of nitrogen application rate, and yield decreased with high nitrogen

application rate [19]. However, the current research and application of varieties mainly concentrated on dry stubble strong and medium gluten wheat and rice stubble medium and weak gluten wheat, it is believed that reasonable fertilization can promote the simultaneous increase of wheat grain yield and protein [20–22]. The suitable nitrogen application amount varies with regional ecological conditions and soil fertility, and the target yield and quality index requirements are also different [23]. Some other studies believed that the increase of nitrogen application rate could increase the spike number, spikelet per spike and wet gluten content of wheat, but too high nitrogen application rate would lead to the decrease of crude protein content and precipitation value of weak gluten wheat in dry stubble [24]. The biological yield of wheat was increased by increasing soil N level and Mo level. Although the biological yield of wheat could be increased significantly with only nitrogen fertilizer application, the economic coefficient decreased. After molybdenum application, the economic coefficient of wheat was significantly improved, dry matter could be fully utilized, and dry matter transfer and accumulation in wheat body to harvest site could be promoted, so as to obtain high yield [7].

In this study, pot experiments were conducted using two wheat varieties grown under different N application rates, Mo application rates. We used cluster analysis, and various agronomic traits were analyzed using analysis of variance (ANOVA). The objectives of this study were: 1) to compare grain yield, nitrogen use efficiency (NUE) and nitrogen-metabolizing enzymes under different N and Mo input rates, and 2) to identify the effects of the interaction between N and Mo on N metabolism, N use efficiency and yield in wheat.

## Materials and methods

### Planting and cultivation management of wheat

Pot experiments were conducted in the experimental farm at Yangtze University (latitude: 30°23'; longitude: 112°29'), Jing-zhou, in 2022. The soil samples from the field's 0~30 cm plough layer in the experimental area before sowing were analyzed.

The soil was calcareous alluvial with pH 6.5, 13.7 g kg$^{-1}$ organic matter, 617.2 mg kg$^{-1}$ alkali-hydrolysable N, 14.5 mg kg$^{-1}$ available P, and 94.2 mg kg$^{-1}$ available K. Then, 2.3 kg of sieved dry soil containing 1.5 g of compound fertilizer (N+P$_2$O$_5$+K$_2$O≥48%), and 30.0 g of organic fertilizer was mixed evenly for each pot (28 cm high, 13 cm in diameter). N was supplied as urea at 0, 75, or 150 kg N ha$^{-1}$ equivalent (N0, N1, N2). Mo was supplied as sodium molybdate (Na$_2$MoO$_4$·2H$_2$O) at 0, 0.75, or 1.5 kg Mo ha$^{-1}$ (Mo1, Mo2, Mo3), applied as basal fertilizer before sowing and thoroughly mixed into the soil.

Five wheat seeds were sown on 17 October 2022, and three seedlings with similar plant sizes were retained at the three-leaf stage. Each pot was top-dressed with 0.2 g of urea at the middle-tillering stage. Crop management followed standard cultural practices. Insects were intensively controlled with pesticides to avoid biomass and yield losses.

### Experimental design

Two commercial wheat cultivars, i.e., Jimai 44 (JM44) and Shannong 23 (SN23), were used in the experiment. The two wheat cultivars used were JM44 (medium-gluten winter wheat, widely cultivated in northern China) and SN23 (high-gluten winter wheat, commonly used in bread-making). These cultivars differ in yield potential and nitrogen use efficiency traits. Three N input, 0, 75, and 150 kg ha$^{-1}$, and three Mo input, 0, 0.75, 1.5 kg ha$^{-1}$ were used as test materials. Nitrogen is applied as urea, Mo was applied as (NH$_4$)$_2$MoO$_4$. The experiment was conducted in a completely randomized design with nine treatments (Table 1).

### Sampling and measurement

**Grain yield and yield components.** Spike number was measured in 3 unsampled pots at wheat maturation stage, and the number of grains per ear and 1000-grain weight of 30 wheat ears were investigated randomly. After all the wheat ears were cut manually, they were threshed by a small seed thresher, dried and weighed, and the grain yield with water content of 13% was calculated.

**Table 1. Total dry weight at maturity, nitrogen accumulation, and nitrogen use efficiency (NUE) relative to N and Mo application rate.**

| | Treatment | TDW (kg ha⁻¹) | NAE (kg/kg) | NRE (%) | PEPN (kg/kg) | N accumulation (kg ha⁻¹) |
|---|---|---|---|---|---|---|
| JM44 | N0+Mo1 | 4821.2 i | | | | 57.6 d |
| | N0+Mo2 | 5228.8 h | | | | 57.6 d |
| | N0+Mo3 | 6339.2 g | | | | 57.6 d |
| | N1+Mo1 | 9969.6 f | 30.2 a | 65.3 a | 77.8 c | 106.5 c |
| | N1+Mo2 | 10837.6 e | 28.4 b | 70.2 a | 83.3 b | 110.3 c |
| | N1+Mo3 | 12615 b | 25.9 c | 66.0 a | 85.3 a | 107.1 c |
| | N2+Mo1 | 11917.8 d | 17.6 d | 62.9 a | 41.4 f | 152.0 ab |
| | N2+Mo2 | 12265.2 c | 15.4 e | 59.1 a | 42.9 e | 146.1 b |
| | N2+Mo3 | 13655.3 a | 14.7 e | 66.1 a | 44.3 d | 156.7 a |
| SN23 | N0+Mo1 | 5453.0 i | | | | 64.5 d |
| | N0+Mo2 | 5722.0 h | | | | 65.2 d |
| | N0+Mo3 | 6207.4 g | | | | 63.8 d |
| | N1+Mo1 | 10791.1 f | 36.3 a | 81.1 b | 80.2 c | 125.3 c |
| | N1+Mo2 | 12534.4 e | 32.0 b | 89.4 a | 85.0 b | 132.2 b |
| | N1+Mo3 | 14041.7 b | 30.5 c | 86.0 ab | 88.4 a | 128.3 c |
| | N2+Mo1 | 13690.2 c | 21.9 d | 65.3 c | 43.9 d | 162.4 a |
| | N2+Mo2 | 13412.3 d | 17.5 e | 62.8 c | 44.1 d | 159.4 a |
| | N2+Mo3 | 15235.0 a | 16.5 f | 64.7 c | 45.5 d | 160.9 a |
| ANOVA | Variety (V) | 35263.1** | 1692.6** | 63.6** | 804.6** | 638.5** |
| | N rate (N) | 714992.0** | 20828.8** | 106.0** | 309007.8** | 12948.2** |
| | Mo rate (M) | 41154.6** | 871.3** | ns | 1697.9** | ns |
| | N*V | 4219.2** | 116.6** | 45.0** | 32.5** | 76.2** |
| | M*V | 69.2** | 65.0** | ns | 18.7** | 4.9* |
| | N*M | 3664.8** | 13.8** | 5.9** | 554.5** | 15.4** |
| | N*M*V | 801.0** | ns | ns | 17.7** | 3.3* |

Different lowercase letters within columns indicate significant differences at P<0.05 across N and Mo rates (n=3). Different lowercase letters within columns indicate significant differences among varieties at P<0.05 (n=3). *P<0.05. **P<0.01. ns, not significant.

**Leaf area index and photosynthetic capacity.** At 7, 14, 21 and 28 days after flowering, 15 plants with uniform growth were selected for each treatment. The LAI was determined by CI-203 leaf area instrument (US CID).

The photosynthetic rate was determined by Li-6800 portable photosynthesis measurement system (Li-Cor), using open gas path, $CO_2$ concentration was about 3μmol L⁻¹, red and blue light source leaf chamber was selected. The photosynthetically active radiation (PAR) was set as 1000 μmol m⁻² s⁻¹. The measurement time was 9:00–11:00 in the morning.

**Dry matter accumulation and nitrogen use efficiency.** At 7d, 14d, 21d and 28d after flowering, 6-point plants with uniform growth were selected for each treatment, washed and defoliated at 105°C for 20 min, dried at 80°C to constant mass before weighing, and then powdered. Samples of the crushed dry matter were taken, weighed at 0.5g for each sample and placed in a Kjeldahl flask, followed by adding 0.3g of potassium sulphate-copper sulphate contact agent. Concentrated Sulfuric acid 2.0mL, digestion furnace digestion. The nitrogen content was measured by automatic Kjeldahl nitrometer and nitrogen uptake was calculated

$$\text{NAE}\left(\text{Kg kg}^{-1}\right) = \frac{\text{Grain yeild with N } - \text{ Grain yield without N}}{\text{N applied}}$$

$$NRE(\%) = \frac{\text{Total N in plant with N} - \text{Total N without N}}{\text{N applied}} \times 100$$

$$PEPN\left(kg\ kg^{-1}\right) = \frac{\text{Grain yield}}{\text{N applied}}$$

**Determination of the nitrogen-metabolizing enzymes.** The nitrate reductase (NR, EC 1.6.6.1) activity assay followed the method proposed by Yu et al [25]. and Imran et al. [26], while GS and GOGAT activities were measured according to Hou et al.[27] and Zhang et al.[28]. We mixed 1.3 mM EDTA and 10 mM cysteine to obtain 4 mL of cold 25 mM sodium phosphate (pH 8.7) buffer. The frozen plant leaf samples were ground into a homogenate in the buffer, which was centrifuged at 4000×g rpm and 4 °C for 30 min. The centrifugation supernatant, 0.1 M $KNO_3$ and 2.82 mM NADH formed the reaction solution. We added NADH to start the reaction, then incubated the mixture for 30 min. The reaction was ended with 1% sulfanilamide and 0.02% N-phenyl-2-naphthylamine and left for 15 min. After centrifugation at 4000×g rpm for 5 min, the absorbance was determined at 540 nm.

The glutamine synthetase (GS) activity was determined according to the methods of (Hou et al., 2019; Zhang et al., 2022) [27–28]. Fresh leaves (1.0 g) were homogenized in 3.0 ml of Tris-HCl (0.05 M, pH 8.0) in a pre-cooled mortar in an ice bath, followed by extraction with $MgSO_4$ (2 mM), DTT (2 mM) and sucrose (0.4 M). The homogenate was centrifuged at 15000 rpm at 4 °C for 20 min. An aliquot of the crude enzyme solution (0.7 ml) and 0.7 ml of 40 mM ATP were added to 1.6 ml of reaction mixture B containing imidazole-HCl buffer (0.1 M, pH 7.4), sodium hydrogen glutamate (20 mM), $MgSO_4$ (80 mM), Cysteine (20 mM), EDTA (2 mM) and hydroxylamine hydrochloride (80 M). For the control, reaction mixture A without hydroxylamine hydrochloride was used instead of reaction mixture B. The mixture was incubated at 25 °C for 15 min. Approximately 1 ml of chromogenic agent (TCA (0.2 M), $FeCl_3$ (0.37 M), HCl (0.6 M)) was added to terminate the reaction. After centrifugation at 5000 rpm for 10 min, the absorbance of the supernatant was measured at 540 nm.

The glutamate synthase (GOGAT) activity was determined according to the methods of (Hou et al., 2019; Zhang et al., 2022) [27–28]. Fresh leaves (0.4 g) were ground in a chilled mortar in 25 mmol/L Tris-HCl buffer (pH 7.6) and the homogenate was centrifuged at 13000 rpm/min for 25 min. The GOGAT reaction mixture consisted of 0.4 ml 20 mmol/L L-glutamine, 0.5 ml 20 mmol/Lα-ketoglutarate, 0.1 ml 10 mmol/L KCl, 0.2 ml 3 mmol/L NADH and 0.3 ml enzyme extract. Through monitoring the oxidation of NADH, GOGAT activity was assayed at 340 nm using a 752UV-Vis spectrophotometer.

## Data analysis

Data were analyzed using three-way analysis of variance with R 4.3.1 (Analytical Software, Tallahassee, FL, USA). The means were compared between N treatments and rice cultivars based on the least significant difference (LSD) test at a 0.05 probability level.

## Results

### Yield and yield components

N and Mo application rate significantly (P<0.01) affected grain yield (Fig 1). Grian yield significantly increased with the increasing of N and Mo input. Average grain yield increases were 61.78% under the different N input, and 11.71% under the Mo input. Grain yield of N1+Mo2 and N1+Mo3 was 6249.9 and 6394.0 kg ha⁻¹, which was 0.68% and 3.0% higher than N2+Mo1, respectively.

Yield component responses to plant density varied with N application rate (Fig 1). spike number significantly (P<0.05) increased with N application rate. spike number in the treatments receiving N was 73.9% and 99.3% higher, respectively,

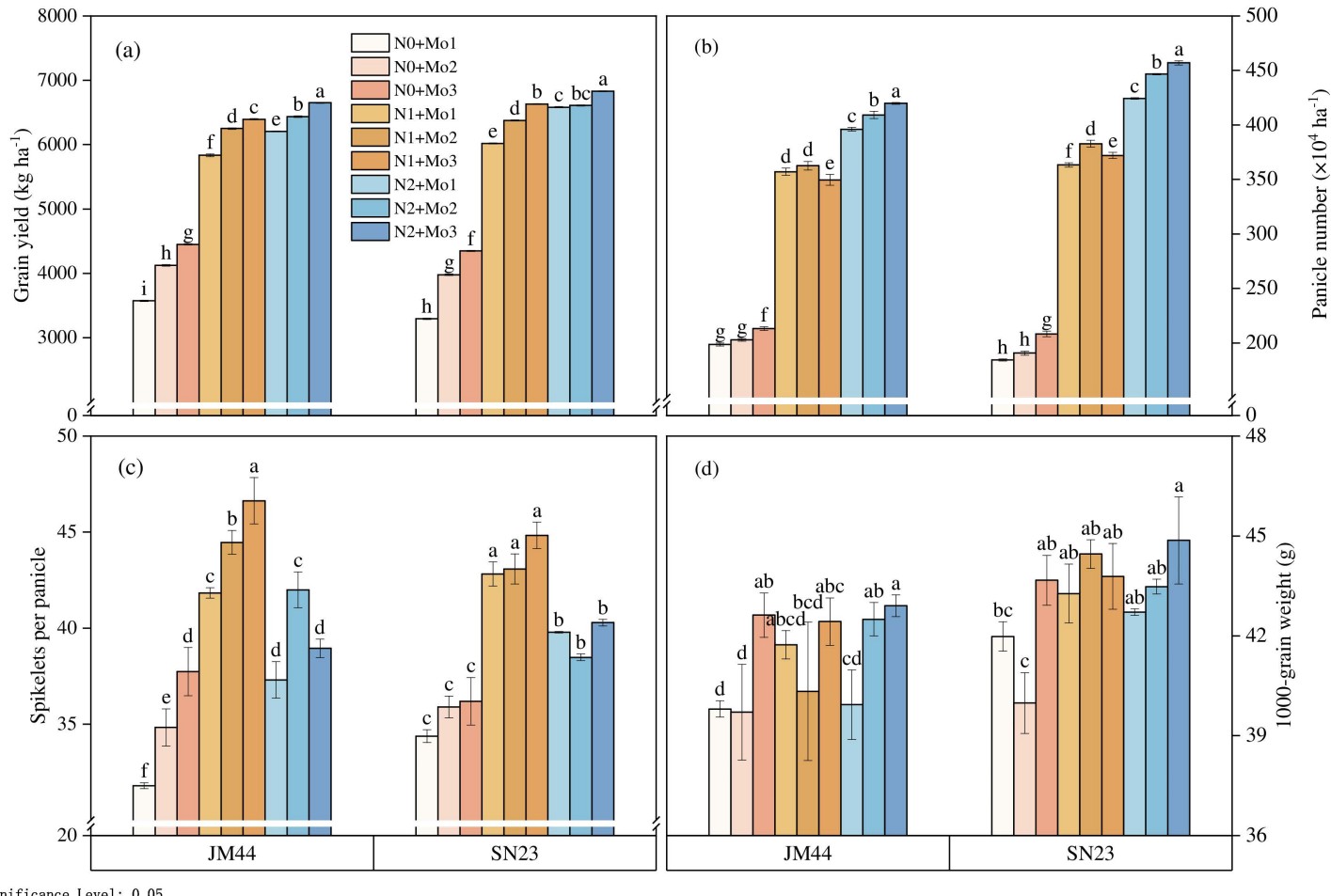

Significance Level: 0.05

**Fig 1. Wheat yield and its components under different N and Mo application levels. (n = 3).**

relative to the N0 treatment, but had not siginifically under the Mo input. Spikelets per spike significant (P < 0.01) different under different N and Mo application rate. The spikelets per spike significantly increased at N1, and either remained constant N2 or decreased at higher N. With the increasing of Mo input, the spikelets per spike of Mo2 and Mo3 were 5.02% and 7.62% higher than Mo1, respectively. The most spikelets per spike was 46.6, which was under N1 + Mo3 treatment. The effects of Mo application rate on 1000-grain weight were significant different (P < 0.05). Average 1000-grain weight in Mo3 increases were 4.41% higher than Mo1.

## Leaf area index and photosynthetic rate

N and Mo application rate significantly (P < 0.01) affected LAI (Fig 2). LAI in the treatments receiving N was 4.83% higher, respectively, relative to the N0 treatment. With the passage of time, LAI after flowering was significantly decreased under different nitrogen treatments, and was steadily decreased from 7 to 14 days after flowering, but was exponentially decreased from 14 to 28 days after flowering. The jointing–heading and post-anthesis stages were most responsive to Mo supplementation, showing significant increases in LAI and photosynthetic rate, which contributed to prolonged green leaf duration and improved grain filling. With the increase of Mo, the leaf area index increased significantly at all growth stages, and Mo2 and Mo3 increased by 13.68% and 19.33% compared with Mo1, respectively.

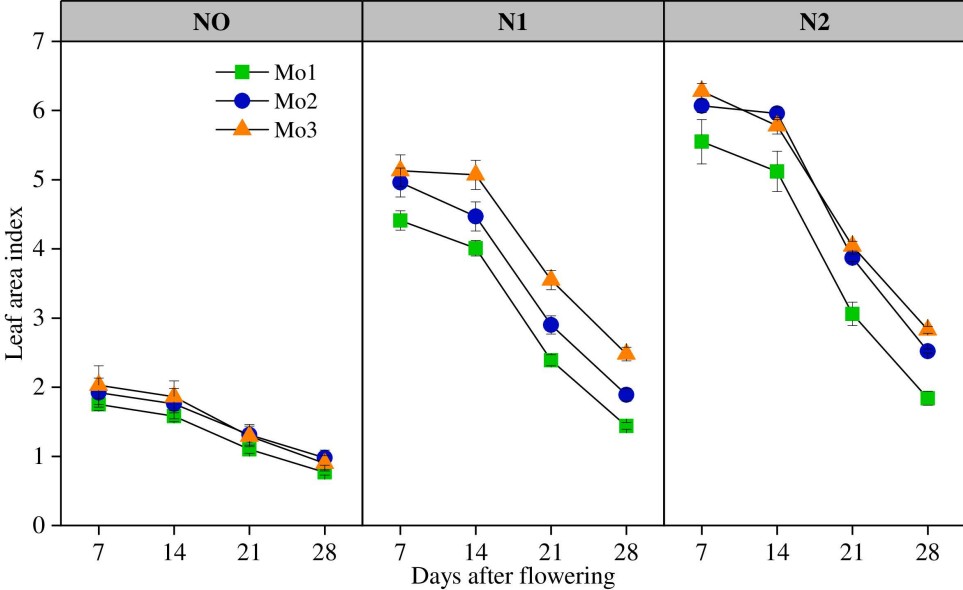

**Fig 2. The variation of leaf area index after flowering under different N and Mo application levels. (n = 6).**

Photosynthetic rate significantly (P < 0.01) different under different Mo input (Fig 3). LAI in Mo2 and Mo3 were 26.47% and 45.07% higher, respectively, relative to the Mo0 treatment. The photosynthetic rate significantly decreased with the passage of growth period, similar to LAI, which decreased steadily from 7-14d and exponentially from 14-28d.

## Total dry weight, nitrogen accumulation, NUE

Total dry weight at maturity significantly (P < 0.05) increased with N and Mo application rate (Table 1). TDW in N1 and N2 were 109.45% and 137.22% higher than N0. TDW significantly increased (N1 + Mo3) and either remained constant (N2 + Mo1/Mo2) or lightly increased (N3) with increasing N and Mo. In contrast, N accumulation significantly increased under different N input, 93.44% and 156.35% higher in N1 and N2 than NO, respectively. The effects of Mo application rate on N accumulation were not significant (P > 0.05).

NUE varied among treatments (Table 1). N1 showed higher NAE, NRE, and PEPN than N2. NAE, NRE and PEPN significantly (P < 0.01) increased at low N rates (N1) but significantly (P < 0.05) decreased at high N rates (N2). Under the N1 treatment, average NAE was 6.13% and 7.13% higher, NRE was 2.24% and 10.62% higher, and PEPN was 19.63% and 20.03% higher in the two wheat varieties compared with N2. NAE and PEPN were significantly (P < 0.01) different among different Mo input levels. NAE significantly decreased with increasing Mo input. Conversely, PEPN increased with increasing Mo input, and the N1 + Mo3 had the highest PEPN.

## Nitrogen metabolizing enzymes

N application significantly (P < 0.01) increased N-metabolizing enzyme activity (Fig 4). Compare with N0, NR, GS and GOGAT increased by 8.94%, 25.95%, and 29.48% in N1 and by 21.92%, 64.21%, and 29.88% in N2, respectively. N-metabolizing enzyme activity significantly improved under different Mo input. Compare with Mo0, NR, GS and GOGAT increased by 1.15%, 4.60%, and 1.95% in N1 and by 3.80%, 7.31%, and 4.94% in N2, respectively. As time went on, N-metabolizing enzyme activity decreased significantly under different nitrogen treatments, stably at 7-14d after anthesis, and exponentially at 14-28d after anthesis.

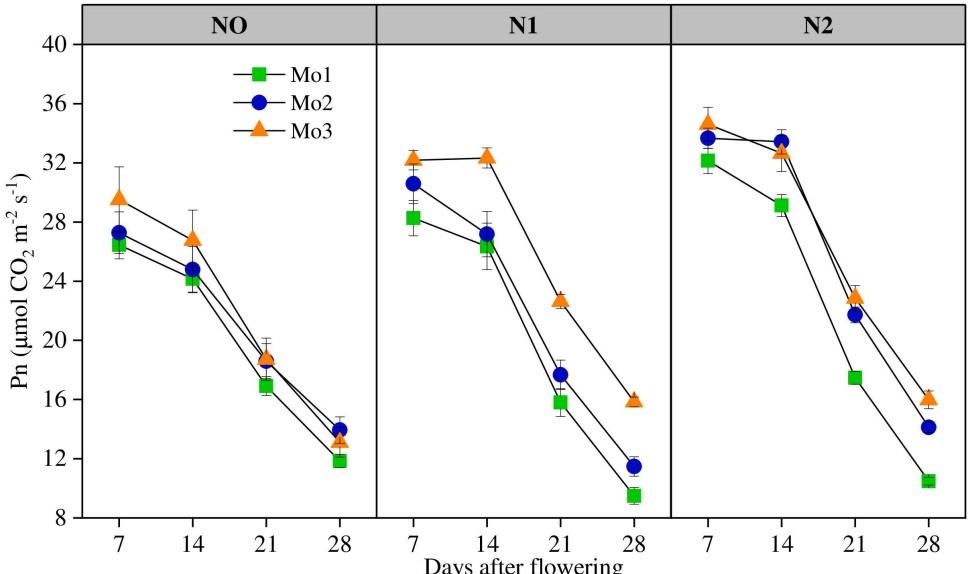

**Fig 3. The variation of photosynthetic rate after flowering under different N and Mo application levels. (n = 6).**

## Correlation analyses of grain yield, yield components, NUE, and nitrogen metabolizing enzymes at various nitrogen and Mo application rates

Correlation matrices among the various grain yield components, nitrogen use efficiency (NUE) parameters, N and Mo application rates are shown in Fig 5. The N application rate was significantly (P < 0.01) and positively correlated with yield, TDW, PN, NR, Naccu, LAI, GOGAT, NRE, and GS. The Mo application rate was significantly (P < 0.05) and positively correlated with GW and Pnco. Additionally, there was a significant (P < 0.01) positive correlation between yield and SP, PNCO, GS, Naccu, LAI, NRE, PEPN, GOGAT, TDW, PN, and NR.

## Discussion

The yield of wheat is mainly affected by three factors: spike number, spikelet per spike and grain weight. Molybdenum plays an irreplaceable role in biological nitrogen fixation and nitric acid reduction [12,29]. Molybdenum is one of the essential trace elements in plants, and the nutritional relationship between molybdenum and nitrogen has attracted great attention [25–26]. The application of molybdenum fertilizer could eliminate the wheat yield caused by high nitrogen application rate and maintain the wheat yield, and whether urea or ammonium nitrate was used as the nitrogen source, molybdenum fertilizer could increase the wheat yield [15,16,30]. Our results indicate a significant N × Mo interaction. While higher N increased yield and enzyme activity, supplemental Mo at low N (N1 + Mo3) achieved yields comparable to high N without Mo (N2 + Mo1). This suggests Mo supplementation can partially offset reduced N input, enhancing NUE. The pot experiment results of Du et al.[7] showed that the yield increase of winter wheat was not stable when molybdenum fertilizer was applied under high nitrogen fertilizer dosage, and nitrogen fertilizer dosage was one of the conditions for molybdenum fertilizer application in winter wheat [31]. We found that both yield and yield components increased with increasing N and Mo application levels, which is consistent with previous conclusions. Under the same N application, with the increase of Mo input, the increase of yield mainly came from the increase of spikelets per spike. With the increasing of Mo input, the spikelets per spike of Mo2 and Mo3 were 5.02% and 7.62% higher than Mo1, respectively. The most spikelets per spike was 46.6, which was under N1 + Mo3 treatment. The effects of Mo application rate on 1000-grain weight were significant

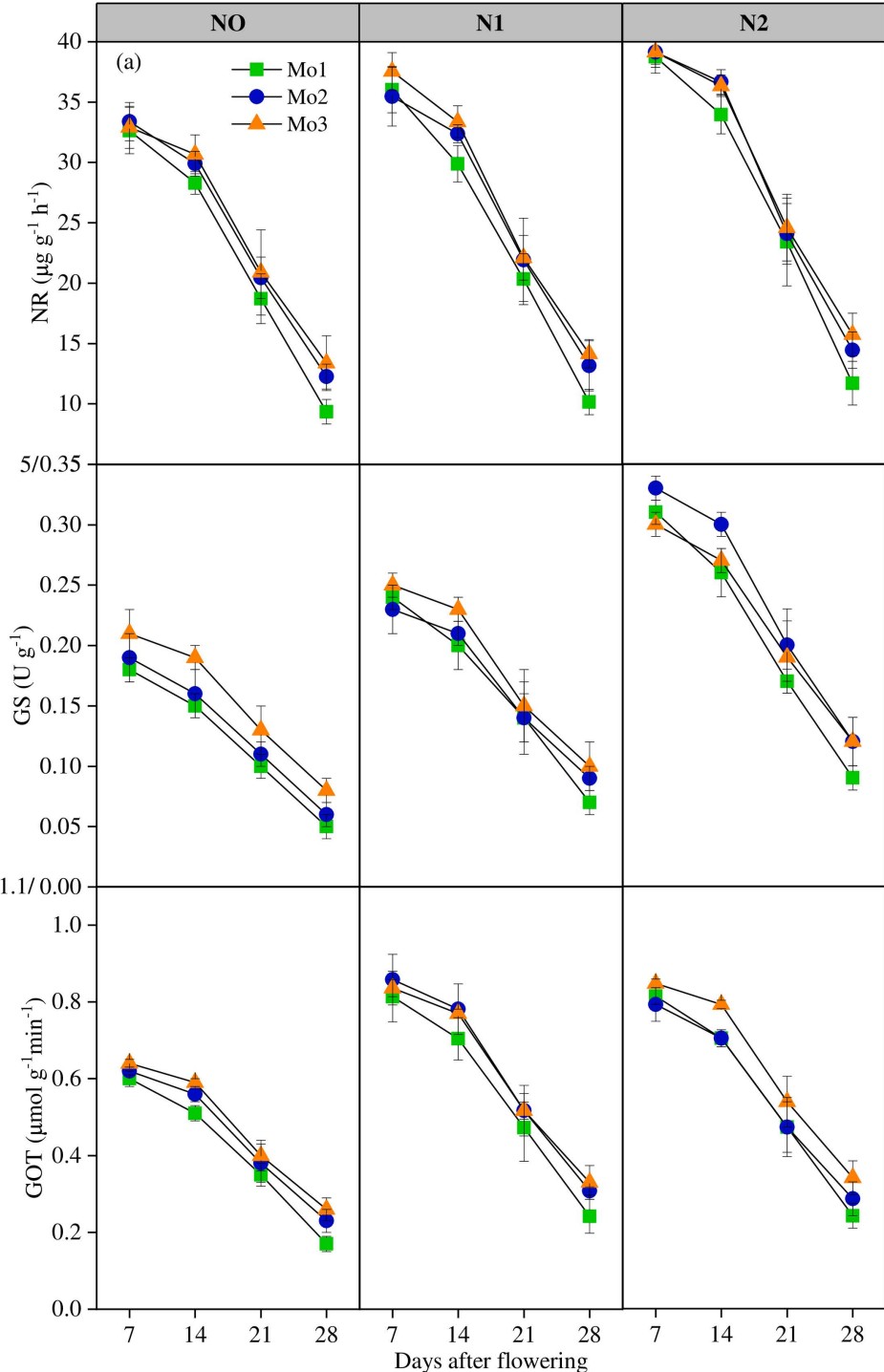

**Fig 4. The variation of N-metabolizing enzymes after flowering under different N and Mo application levels. (n=6).**

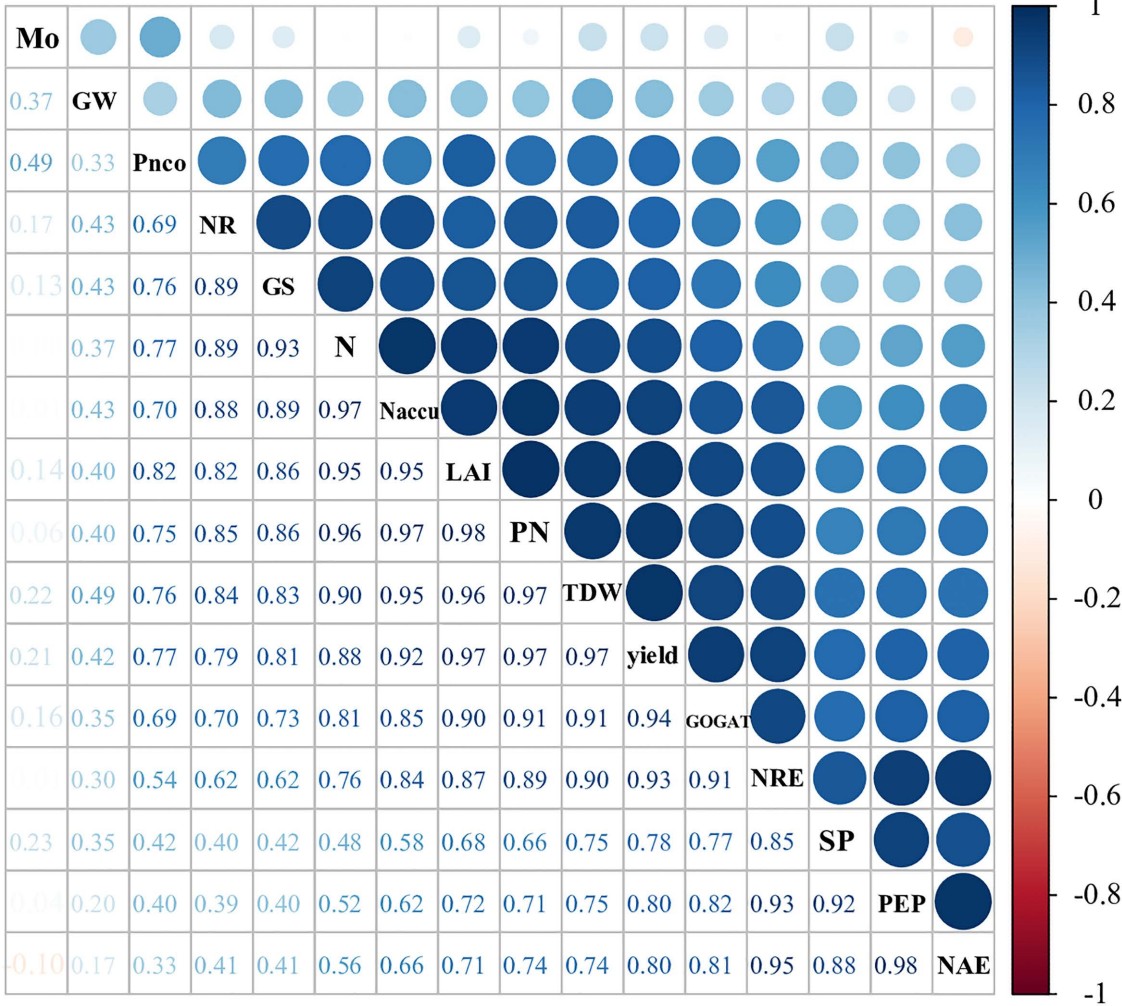

**Fig 5. Correlation matrix of various grain yield and population growth parameters (n = 54).** N, nitrogen application rate; Mo, Mo application rate; PN, spike number m$^{-2}$; SP, spikelets per spike; GW, 1,000-grain weight; TDW, total dry matter; Naccu, nitrogen accumulation; NR, nitrate reductase; GS, gutamine synthetase; GOGAT, NADH-glutamate synthase. Numbers are determination coefficients.

different (P < 0.05). Average 1000-grain weight in Mo3 increases were 4.41% higher than Mo1 while spike number was not significantly different under different Mo input. In this study, grain yield of N1 + Mo2 and N1 + Mo3 was 6249.9 and 6394.0 kg ha$^{-1}$, which was 0.68% and 3.0% higher than N2 + Mo1, respectively. N1 + Mo2 and N1 + Mo3 showed no difference from N3 + M1, indicating that Mo has the potential to reduce the N input and compensate for the yield decline.

Nitrogen is an important element to ensure the high yield of wheat [32]. At present, in order to maintain a high yield, the application rate of nitrogen fertilizer has been maintained at a high level, which is the main reason for the low utilization rate of nitrogen fertilizer [33]. Tian et al.[34] showed that, nitrogen reduction affected the nitrogen uptake by white gluten wheat plants in the dry stubble half winter under the condition of low soil fertility, and the nitrogen apparent use efficiency decreased, but the nitrogen partial productivity and nitrogen physiological efficiency increased, while the nitrogen harvest index was little affected. In our study, NAE, NRE and PFP significantly (P < 0.01) increased low N rates (N1) but significantly (P < 0.05) decreased with at high N rates (N2). NAE significantly decreased with the increased of Mo input.

Conversely, the PEPN increased with the increased of Mo input, the N1+Mo3 had the higher PEPN. At high N rates, NUE decreased likely due to luxury N uptake exceeding the assimilation capacity of NR, GS, and GOGAT, leading to inefficient N utilization and lower NRE and NAE. This study showed excessively increasing the amount of N fertilizer would reduce NAE and NRE, which is consistent with the results of most studies [35–36]. Rational nitrogen fertilizer application is of great significance in improving wheat NUE. PEPN significantly improved under the Mo application rates, and it is significantly increased under low nitrogen (N1) condition, indicating that increasing Mo under low nitrogen (N1) treatment can effectively improve wheat nitrogen use efficiency.

Lack of molybdenum will inhibit the synthesis of photosynthetic pigments and lead to yellowing of plant leaves [35–36]. Appropriate amount of molybdenum is beneficial to plant growth, while excessive molybdenum will lead to crop yield reduction and agricultural product quality decline [25]. Molybdenum application promoted the photosynthesis of winter wheat [36]. Under the condition of soil molybdenum pollution, the chlorophyll content of winter wheat decreased and stomatal restriction led to the decrease of photosynthetic rate, which affected the photosynthetic capacity of winter wheat, and finally led to the significant reduction of biological yield [35]. This is similar to the results of this study. Increasing Mo input significantly increased the LAI and photosynthetic rate at each growth stage, Mo could significantly prolong the duration of green leaf, improve the net photosynthetic rate at the later growth stage, promote grain filling and fruit setting, and increase yield.

NR plays an important role in plant regulation and nitrogen assimilation, and N and protein concentrations in leaves are related to their activities, while NR, GS and GOGAT play a key role in nitrogen metabolism through synergistic interaction with N [36]. In this study, appropriate nitrogen and molybdenum applications significantly increased N-metabolizing enzyme activity. Relevant studies showed that molybdenum application reduced the nitrate nitrogen content of winter wheat varieties [25], promoted the conversion of inorganic nitrogen to organic nitrogen, and provided more nitrogen sources for the synthesis of substances in the subsequent metabolic chain. At the same time, it could avoid the possible poisoning of plants caused by excessive accumulation of nitrate nitrogen [2,5,36,37]. Although chlorophyll content was not directly measured in this study, previous work has shown that Mo supplementation enhances chlorophyll biosynthesis and Rubisco activity, thereby improving photosynthetic efficiency [26,38]. This is similar to the results of this study, molybdenum application can effectively improve nitrogen metabolic pathway, enhance N-metabolizing enzyme activity, extend nitrogen metabolic intensity in later growth period, and improve nitrogen use efficiency, so as to achieve increased production and efficiency.

## Conclusions

This study evaluated the contribution of photosynthetic characteristics, NUE, and nitrogen-metabolizing enzymes to yield under reduced nitrogen input and molybdenum input. The results showed that increasing N and Mo inputs significantly increased yield. Moreover, increasing Mo under the low nitrogen (N1) treatment effectively compensated for the yield loss, even surpassing the yield under the high nitrogen (N2+Mo1) treatment. The yield increase was primarily attributed to higher grain number per spike and TDW, which were associated with enhanced NRE, PEPN, LAI after flowering, photosynthetic rate, and GOGAT activity under increased Mo input. Therefore, increasing Mo input in reducing nitrogen application rates could effectively reduce yield loss and improve NUE. These findings have important implications for achieving high yield and high nutrient-use efficiency in modern wheat-production systems.

## Supporting information

**S1 Data. Minimal dataset underlying the findings reported in this study.** The file contains the raw and processed data on grain yield, total dry weight (TDW), photosynthetic parameters, nitrogen-metabolizing enzyme activities (NR, GS, GOGAT), and nitrogen use efficiency (NUE) indices of winter wheat cultivars under different nitrogen and molybdenum treatments.
(XLSX)

## Author contributions

**Conceptualization:** Youning Wang.

**Funding acquisition:** Youning Wang.

**Investigation:** Youning Wang.

**Methodology:** Youning Wang.

**Project administration:** Youning Wang.

**Validation:** Qixia Wu.

**Visualization:** Di Yang.

**Writing – original draft:** Di Yang.

**Writing – review & editing:** Qixia Wu.

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
