## [Decision Letter · Decision Letter 0]

19 Mar 2025

Dear Dr. Wang,

Thank you for submitting your manuscript to PLOS ONE. After careful consideration, we feel that it has merit but does not fully meet PLOS ONE’s publication criteria as it currently stands. Therefore, we invite you to submit a revised version of the manuscript that addresses the points raised during the review process.

We look forward to receiving your revised manuscript.

Kind regards,

Nabin Rawal, PhD

Academic Editor

PLOS ONE

“This work was supported by the Natural Science Funds of Hubei Province of China(2024AFB1015).”

Additional Editor Comments:

Please consider the enclosed reviewer's comments and make the necessary changes to your manuscript based on their advice/ suggestion and send the revised manuscript for further processing so that we can resend to the reviewers.

Reviewers' comments:

Reviewer's Responses to Questions

**Comments to the Author**

1. Is the manuscript technically sound, and do the data support the conclusions?

Reviewer #1: Partly

Reviewer #2: Yes

Reviewer #3: Partly

Reviewer #4: Yes

2. Has the statistical analysis been performed appropriately and rigorously?

Reviewer #1: Yes

Reviewer #2: Yes

Reviewer #3: Yes

Reviewer #4: Yes

3. Have the authors made all data underlying the findings in their manuscript fully available?

Reviewer #1: Yes

Reviewer #2: Yes

Reviewer #3: Yes

Reviewer #4: Yes

4. Is the manuscript presented in an intelligible fashion and written in standard English?

Reviewer #1: Yes

Reviewer #2: Yes

Reviewer #3: No

Reviewer #4: Yes

Reviewer #1: Specific comments:

1. Please recheck the abbreviations that have been mentioned in the whole manuscript. Please elaborate on them at least once in the first place in the abstract and other sections of the manuscript.

In the abstract, authors are asked to revise some typing mistakes.

2. The figures in the results section are of good quality, but sometimes the fonts are so small that they are hard to read. Also, the patterns in the columns of each figure should be changed to other patterns or colors that are easy to read.

3. The introduction section is well written, and in my opinion, it sheds light on the problem in a concise manner. Overall, the objectives are briefly explained and are good to go with.

4. The materials and methods section is nicely presented and well described. Some references are missing.

5. Please check the whole manuscript; the format should be improved, such as there are some unit presentations. Put the relevant unit with each value across the manuscript.

Please add some recent literature to discuss your results in the discussion section.

6. Please rewrite the conclusion. The conclusion is recommended to be supported by the data shown in tables, put detail of any limitations of this study, describe implications of this study, and provide recommendations for future perspectives.

7. The references in the text as well as in the list should be formatted according to the style of the journal.

Main Comments:

1. Consider clarifying the specific mechanisms by which regulated N and Mo management might influence NUE and yield.

2. Could you provide a more detailed explanation of how these increases compare to previous studies or expected outcomes?

3. It would be helpful to specify which growth stages were most affected and why.

4. Consider discussing any potential trade-offs or limitations of this approach, such as economic feasibility or environmental concerns.

5. Adding data on how Mo influences photosynthetic enzyme activity or chlorophyll content could strengthen this statement.

6. Consider discussing the potential economic and environmental implications of this fertilizer management strategy.

7. It might be helpful to discuss possible physiological explanations for why high N rates decrease these efficiencies.

Reviewer #2: Dear Authors,

The subject of the study is interesting and topical, with scientific and practical importance.

The following aspects are brought to the attention of the authors.

1.

Page 1, Line 13

“the effects” instead of “theeffects”

2.

Line 14

"Se", what does it mean?

“Se application levels”

3.

Line 26

"Mo0", what does it represent?

"Mo1", "Mo2" and "Mo3" (Lines 14 - 15) have been explained, but not "Mo0".

Clear presentation of treatments is necessary.

4.

Lines 80 - 81

It is recommended to clarify certain terms, with reference to wheat

e.g.

"spikelet per panicle" and wet gluten content of wheat

5.

Lines 136 – 138

The NAE formula (lines 136 - 138) is appropriate to be presented as an equation, and not in text format.

Similar recommendations for "NRE" and for "PFPN", lines 138 – 142

6.

Numbering of bibliographical sources in the text

e.g. Line 154

“Hou et al., 2019; Zhang et al., 2022”

Similar to Line 167

7

Line 163

“FeCl3” instead of “FeCl3”

8.

It is recommended to revise the term "panicle" in reference to wheat.

In several places in the text, throughout the article, the term is used.

Revision and correction are recommended, with appropriate terms for wheat.

9.

References

Some bibliographic sources have blank spaces in the text line.

Revision and correction are required.

Some journal titles are written in full text, while others are written in abbreviated form.

Proofreading and unitary writing are recommended, in accordance with the Instructions for Authors, PLOS One journal

Italic Font Style for species name

e.g.

Line 388

“Triticum aestivum L.”

10.

Figure 5, Line 439

“m-2” instead of “m-2”

11.

Figure 5, Line 439

“*” and “**” are explained in the figure title, but are not found in the figure, associated with the obtained values

Reviewer #3: In this study, authors analyzed the grain yield, NUE parameters, and N metabolic enzyme activities in wheat under the combined application of N and Mo using two cultivars. Results revealed that high Mo and low N input is beneficial for yield production, as it can prolong the duration of green leaf, improve the activity of N metabolic enzymes in middle and late growth. Several suggestions and questions are as follows:

1. The language needs to be improved by the native English speakers, as many places are not following the scientific English writing standards and difficult to understand. I do not want to point these places one by one.

2. Line 13, there should be a blank space in “theeffects”.

3. Line 14, Se? It is Mo.

4. Line 26-27, the activities of NR, GS, and GOGAT. How N1 and N2 can compare with Mo0?

5. How to conduct the N and Mo treatment? It is not clear. The materials have not been described in the M&M section.

6. Line 109-113, what is this? I doubt whether this manuscript has been written by the authors?

7. Line 136, authors described the NAE as nitrogen agronomic efficiency, but in abstract, NAE is referred as nitrogen absorption efficiency.

8. Line 141-142, “N fertilizer (PFPN, kg kg–1) = [grain yield / amount of N fertilizer applied] “?

9. Line 190, what is “N3”?

10. Authors performed this experiment using two cultivars, but they did not refer these cultivars in the manuscript. They should introduce the background of these two cultivars, describe the results of these two cultivars, and discuss the similarities and the differences of these two cultivars.

11. The interaction between N and Mo needs to be further discussed.

Reviewer #4: The following corrections should be made to the article:

1-In line 13 the effects

2- In line 14 It should be corrected to Se instead of Mo

3- Inl line 77 (20-22) There is no reference number 21.

4-In line 89 It should be corrected to field experiments instead of pot experiments

5- In line 90 plant densities There is no experiments plan with plant density in the article.

6-In line 98 2022 instead 2021

7-In line 163 FeCI3 (The number 3 should be written as a subscript)

8-In line 168 HCl

9- Inline 210 ve 216 Table 2 instead of table 1

**Do you want your identity to be public for this peer review?** For information about this choice, including consent withdrawal, please see our Privacy Policy

Reviewer #1: **Yes:** Muhammad Usman

Reviewer #2: No

Reviewer #3: No

Reviewer #4: No

---

## [Author Response · Author response to Decision Letter 1]

23 Sep 2025

Dear [Editor],

We appreciate the reviewers' thoughtful and constructive comments on our manuscript. In response to their suggestions, we have improved the clarity and readability of the manuscript, enhanced the discussion of the results, and corrected various typographical and formatting issues. We have also expanded the conclusion to discuss the study's limitations and future research directions. Additionally, we have provided further context for the wheat cultivars used in the study and clarified abbreviations and terminology. We believe these revisions significantly improve the quality of the manuscript and address all concerns raised by the reviewers.

We look forward to your feedback and the opportunity to proceed with publication.

Sincerely,

[Youning Wang]

---

## [Decision Letter · Decision Letter 1]

14 Dec 2025

Effects of increasing molybdenum under low nitrogen input on yield, nitrogen-metabolizing enzymes and nitrogen use efficiency of winter wheat cultivars

PONE-D-25-09819R1

Dear Dr. Wang,

We’re pleased to inform you that your manuscript has been judged scientifically suitable for publication and will be formally accepted for publication once it meets all outstanding technical requirements.

Kind regards,

Nabin Rawal, PhD

Academic Editor

PLOS One

Additional Editor Comments (optional):

We truly appreciate your consideration to publish your research findings with the PLOS One. Thank you for addressing the comments and suggestion from subject matter experts (four reviewers), and I would like to inform you that it may be proceeded for further processing.

Reviewers' comments:

Reviewer's Responses to Questions

**Comments to the Author**

Reviewer #1: All comments have been addressed

Reviewer #3: All comments have been addressed

2. Is the manuscript technically sound, and do the data support the conclusions?

Reviewer #1: Yes

Reviewer #3: Yes

3. Has the statistical analysis been performed appropriately and rigorously?

Reviewer #1: Yes

Reviewer #3: Yes

4. Have the authors made all data underlying the findings in their manuscript fully available?

Reviewer #1: Yes

Reviewer #3: Yes

5. Is the manuscript presented in an intelligible fashion and written in standard English?

Reviewer #1: Yes

Reviewer #3: Yes

Reviewer #1: All comments have been addressed, and the manuscript is recommended for acceptance for publication. Best of luck!

Reviewer #3: (No Response)

**Do you want your identity to be public for this peer review?** For information about this choice, including consent withdrawal, please see our Privacy Policy

Reviewer #1: No

Reviewer #3: **Yes:** Hongmei Cai

---

## [Editor Report · Acceptance letter]

PONE-D-25-09819R1

PLOS One

Dear Dr. Wang,

I'm pleased to inform you that your manuscript has been deemed suitable for publication in PLOS One. Congratulations! Your manuscript is now being handed over to our production team.

Kind regards,

on behalf of

Dr. Nabin Rawal

Academic Editor

PLOS One